# VulFinder: A Multi-Agent-Driven Test Generation Framework for Guiding Vulnerability Reachability Analysis

## Abstract

Reusing third-party components in the software supply chain (SSC) may introduce risks of vulnerabilities. After disclosing a new third-party component vulnerability, developers need to determine whether the project is affected by the specific vulnerability, which requires vast manpower and resources for assessment. Current approaches mainly rely on dependency-based tools and genetic algorithm-based methods to assess the reachability problem of vulnerabilities in SSC. However, these methods suffer from several issues: they ignore the actual invocation of the vulnerable code, resulting in high false positive rates, are limited to certain vulnerabilities, leading to high false negative rates, and are confined to the Java ecosystem. To overcome these challenges, we propose VulFinder, a multi-agent driven framework for validating vulnerability reachability. VulFinder begins by using static code analysis tools to construct function call paths between downstream applications and dependency vulnerability APIs. Leveraging a multi-agent mechanism comprising a distillator, discriminator, generator, and validator, VulFinder iteratively generates exploit tests for methods along the call graph, effectively validating vulnerability reachability by executing these tests on downstream applications. By integrating the code comprehension capabilities of large language models (LLMs) with the multi-agent framework, VulFinder addresses the coverage limitations of existing tools, reduces false alarms and missed alarms, and demonstrates robust generalizability across multiple programming languages. Experiments show that VulFinder achieves 21% accuracy improvement over the state-of-the-art tool VESTA and 7% accuracy improvement over the popular baseline tool TRANSFER on the Java dataset and also demonstrates robust generalizability on the Python dataset, significantly reducing false positives and false negatives and delivering an average efficiency improvement of more than 1.5×.

## 1 Introduction

Modern software development frequently incorporates third-party open-source components to streamline the development process. However, these third-party open-source components may contain vulnerabilities, exposing downstream applications in the software supply chain (SSC) to security risks stemming from weaknesses Bavota et al. (2015); Chen et al. (2020). A notable example is the Log4j vulnerability disclosed in 2021 CSRB (2022), which enabled attackers to execute remote code by crafting malicious log messages wired (2021). Addressing this challenge has become critical for software developers and organizations, as they must effectively assess and verify whether vulnerabilities in dependent components are exploitable in practice, i.e., vulnerability reachability analysis.

To assess vulnerability reachability, researchers have developed various tools for determining whether vulnerabilities in SSC are reachable, which can be broadly categorized into dependency-based and heuristic-based approaches. Dependency-based approaches utilize vulnerability databases to identify vulnerable dependencies within the upstream dependency network. Tools like GitHub Dependabot He et al. (2023) and OWASP Dependency-Check OWASP (2024) alert users when a vulnerable dependency is detected and recommend updating to a secure version. However, these methods typically overlook the actual invocation of vulnerabilities within the specific codebase, leading

to high false positive rates. For instance, as shown in Figure 1, the vulnerability TwelveMonkeys-595 Snyk (2021a) can cause a program crash when a user imports TwelveMonkeys and uses the *ImageIO.read()* method to process a non-standard JPEG file. Although the user imports the vulnerable version of TwelveMonkeys, it is practically unreachable as it lacks a call path to the vulnerable code. Dependency-based tools like GitHub Dependabot incorrectly flag this dependency as risky, leading to false positives and unnecessary efforts by developers to switch versions or fix code.

Heuristic-based approaches, such as SIEGE Iannone et al. (2021) and TRANSFER Kang et al. (2022), employ genetic algorithms combined with vulnerability knowledge to generate test cases for downstream applications. These tools assess vulnerability reachability by executing exploit tests. However, they rely heavily on manually defined rules, which limit their coverage and effectiveness. As shown in Figure 1, *density-converter* patrickfav (2016) depends on TwelveMonkeys, and it does invoke the vulnerable code of TwelveMonkeys-595, i.e., vulnerability is actually reachable. However, since TwelveMonkeys registers its image processing implementation with the Java standard *ImageIO* framework via the Java Service Provider Interface (SPI) mechanism, existing heuristic-based tools like TRANSFER fail to cover such mechanisms. This results in missed detection of the vulnerability's reachability, potentially allowing developers to overlook critical risks.

To efficiently verify the reachability of vulnerabilities with SSC to downstream applications, this study introduces VulFinder, a vulnerability reachability verification method based on exploit test generation. Unlike existing heuristics-based tools such as SIEGE and TRANSFER, VulFinder does not rely on manually defined domain knowledge, achieving improved coverage while demonstrating cross-language generalization capabilities. Furthermore, VulFinder incorporates a multi-agent mechanism to generate vulnerability exploitation tests and perform discrimination and verification. By combining dynamic test program execution, VulFinder effectively reduces false alarms and missed alarms in vulnerability reachability analysis.

Specifically, VulFinder begins by constructing a function call path that links the downstream application to the vulnerable module of the dependency. For each function along the call path, VulFinder iteratively generates the corresponding vulnerability exploit tests using a multi-agent mechanism comprising a distillator, discriminator, generator, and validator. The distillator implements auto-prompting, which is able to refine redundant user inputs into concise prompts that are suitable for the generation of exploit tests. The discriminator, generator, and validator collaboratively assess vulnerability reachability, generate relevant test programs, and validate their effectiveness. By evaluating reachability and verifying the generated results, the discriminator and validator are able to minimize false positives and false negatives. Ultimately, VulFinder produces a test program for the specified method in the downstream application, enabling efficient verification of the vulnerability reachability.

We evaluate the effectiveness and efficiency of VulFinder on Java and Python datasets. Compared to the state-of-the-art tool, VESTA Chen et al. (2024), VulFinder achieves 21% accuracy improvement on the Java dataset. Compared to the popular baseline tool, TRANSFER, VulFinder demonstrates 7% accuracy improvement on the Java dataset and also demonstrates robust generalizability on the Python dataset, obtaining 33% accuracy improvement, significantly reducing false positives and false negatives, i.e., achieving 33% and 363% Recall improvement respectively, and delivering an average efficiency improvement of more than 1.5 times. Ablation experiments further highlight the contributions of the distillator, discriminator, and validator to VulFinder. A replication package for this work is available at `https://github.com/OpenLabSE/VulFinder`.

## 2 VULFINDER

We propose VulFinder, as shown in Figure 2, which is mainly divided into two modules, i.e., call graph generation and multi-agent driven test generation. Specifically, VulFinder begins by analyzing the static code of downstream software applications and their dependencies to construct call graphs for the vulnerable code modules. It then iteratively generates tests for the functions within the call graph. In each iteration, multiple agents, including the distillator, discriminator, generator, and validator, collaborate to produce valid exploit tests. Finally, users execute the generated vulnerability exploits on the downstream application to check whether the vulnerability can be triggered.

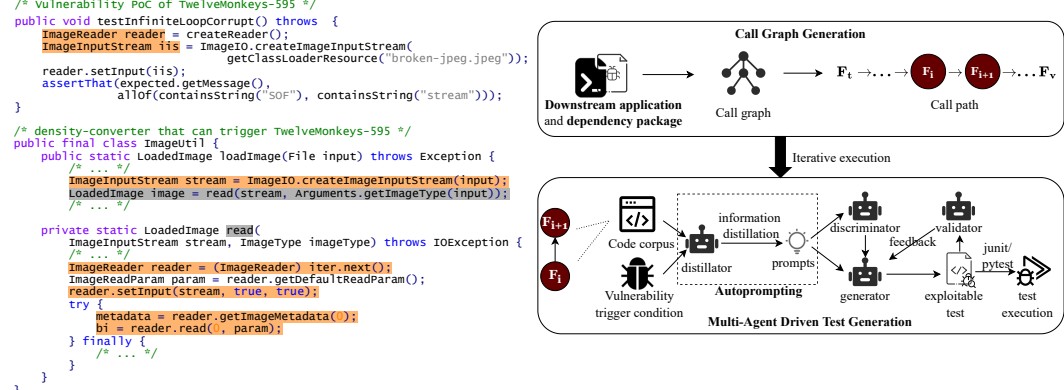

```
108  /* Vulnerability PoC of TwelveMonkeys-595 */
109  public void testInfiniteLoopCorrupt() throws {
         ImageReader reader = createReader();
110      ImageInputStream iis = ImageIO.createImageInputStream(
                         getClassLoaderResource("broken-jpeg.jpeg"));
111      reader.setInput(iis);
         assertThat(expected.getMessage(),
                allOf(containsString("SOF"), containsString("stream")));
     }

112  /* density-converter that can trigger TwelveMonkeys-595 */
     public final class ImageUtil {
113      public static LoadedImage loadImage(File input) throws Exception {
             /* ... */
114          ImageInputStream stream = ImageIO.createImageInputStream(input);
             LoadedImage image = read(stream, Arguments.getImageType(input));
             /* ... */
115      private static LoadedImage read(
             ImageInputStream stream, ImageType imageType) throws IOException {
116          /* ... */
             ImageReader reader = (ImageReader) iter.next();
117          ImageReadParam param = reader.getDefaultReadParam();
             reader.setInput(stream, true, true);
118          try {
                 metadata = reader.getImageMetadata(0);
                 bi = reader.read(0, param);
119          } finally {
                 /* ... */
120          }
         }
     }
```

Figure 1: TwelveMonkeys-595 vulnerability PoC and code examples which trigger the vulnerability in the downstream application (density-converter) that depends on the vulnerability

Figure 2: Overview of proposed VulFinder

## 2.1 CALL GRAPH GENERATION

VulFinder combines static code analysis tools to build call graphs between methods by parsing the source or byte code of the project. This step aims to preliminarily assess the reachability of the vulnerability, with method signatures and input information on the call paths also serving as critical context for generating vulnerability exploits. First, the user needs to input the code corpus of the dependencies and the downstream application, along with the specification of the vulnerable code module. VulFinder then generates the call graph for the vulnerable module in the dependency code and collects the API sequences from the call graph. Next, it constructs the global method call graph for the downstream application and matches calls to dependency APIs using regular expressions. By mapping the downstream application's calls to dependency APIs against the API sequences of the dependency, VulFinder identifies the method call paths connecting the downstream application to the vulnerable APIs. The implementation is carried out in Java and Python projects using javagc-static gousiosg (2017) and pyan davidfraser (2020) respectively. As shown in Figure 2, the call path between the the downstream application's method ($F_t$) and the vulnerable module ($F_v$) in dependency guides VulFinder in iteratively generating vulnerability exploits.

## 2.2 MULTI-AGENT DRIVEN TEST GENERATION

As illustrated in Figure 2, VulFinder begins with the vulnerable module $F_v$ and progressively generates vulnerability exploits for each upstream function along the call path until it reaches the downstream application's method $F_t$. In each iteration, where $F_i$ calls $F_{i+1}$, VulFinder employs a multi-agent-driven framework to generate exploit tests. First, the distillator extracts relevant information from user inputs to create a refined prompt, leveraging the multi-language comprehension capabilities of LLM to eliminate dependence on predefined rules. Next, the discriminator evaluates whether the vulnerability is genuinely reachable and, if so, forwards the result to the generator, which produces the exploit tests to reduce false positives. Finally, the Validator verifies the validity of the generated exploits using a reflection mechanism, further addressing potential hallucinations from the LLM.

**Distillator.** To transform raw user input into prompts suitable for test generation and eliminate redundant information, VulFinder incorporates a distillator to implement autoprompting generation. Unlike existing tools that rely on manually defined coverage rules, the training corpus of LLMs inherently includes a wide range of implementation mechanisms across different programming languages. The distillator leverages this capability to extract relevant information from user input and convert it into a concise prompt. As shown in Figure 3, user inputs typically include **textual details** about the vulnerability, the **vulnerability PoC**, the **method to be tested** within the downstream ap-

**Algorithm 1** Autoprompting Generation

**Input**: useInput, sampleNum
**Output**: promptSet

1: greedyPrompt ←
   $D_a$(useInput, basicPrompt, temperature=0)

2: promptSet ← [greedyPrompt]
3: **while** |promptSet| < sampleNum **do**
4:    prompt ←
      $D_a$(useInput, basicPrompt, temperature=1)
5:    promptSet.append(prompt)
6: **end while**
7: **return** promptSet

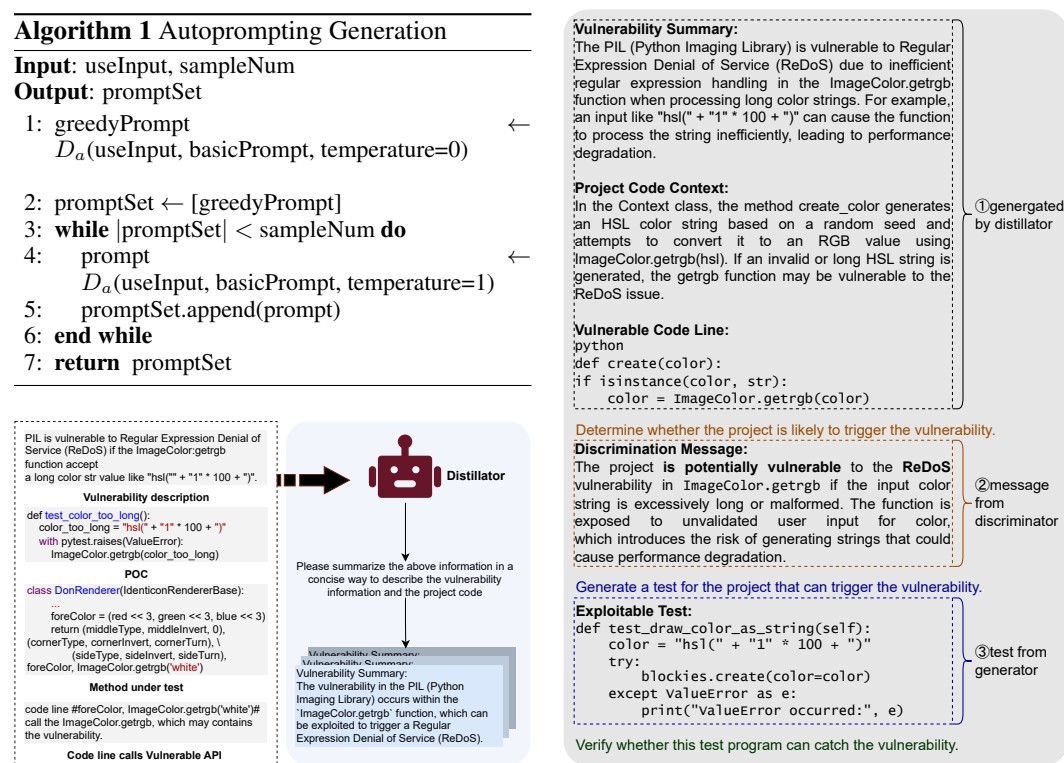

Figure 3: Process of Autoprompting Generation          Figure 4: Example of Prompts

plication, and specific lines of code invoking the vulnerable API. These inputs often contain extraneous code context or irrelevant vulnerability-related details that are not essential for test generation. The Distillator refines and distills the critical information, including the downstream application context, vulnerability details, key code lines, and control flow information within function calls, ensuring the generated prompts are precise and optimized for producing effective exploit tests.

Algorithm 1 details the autoprompting generation process. The algorithm takes the user's context (shown in Figure 3) and a predefined number of samples as input to produce a set of prompts. As illustrated in Figure 3, VulFinder employs a distillator to process the user's input along with a basic instruction: *"Please summarize the above information concisely to describe the vulnerability information and the project code."* This generates streamlined prompts through knowledge distillation of the user-provided context. Using a greedy sampling strategy with a temperature coefficient of 0 (line 1), the algorithm initially generates prompt Chen et al. (2021); Xia & Zhang (2023). Subsequently, it employs higher-temperature sampling (line 4) to produce more diverse prompts. The generated prompts are iteratively added to a prompt set until the set reaches the predefined sample capacity. Figure 4 provides an example of the distilled key information obtained through the autoprompting generation process, including the vulnerability summary, the project code context, and the vulnerable code line. These distilled prompts are utilized by subsequent agents, such as the discriminator, generator, and validator.

**Discriminator.**    For each prompt generated through autoprompting, VulFinder employs a discriminator agent to assess whether the downstream application can trigger the vulnerability. The discriminator's primary role is to filter out cases where the dependency vulnerability cannot be triggered, thereby avoiding unnecessary generation efforts and reducing false positives. As shown in Figure 4, the discriminator processes the concise prompt ① alongside a basic instruction: *"Determine whether the project is likely to trigger the vulnerability."* It then outputs a discrimination result and

a corresponding message ②. If the result is positive, the prompt is enriched with the discrimination message and passed to the generator for further processing.

**Generator.** The generator creates vulnerability exploit tests using the distilled user input and discriminative message. As shown in Figure 4, the generator processes the distilled user input ① and the message from discriminator ②, alongside a basic instruction: *"Generate a test for the project that can trigger the vulnerability"*. In the scenario where the function *create()* calls *ImageColor.getrgb()*, the vulnerability in *ImageColor.getrgb* is triggered when processing an excessively long string input. The distillator provides critical context about the code and the vulnerability triggering condition. Combined with the discriminator's message, the generator can analyze the internal data control flow and deduce the input parameters for the *create()* needed to trigger the vulnerability. It then generates a corresponding test case, which is subsequently passed to the Validator for verification.

**Validator.** The validator mitigates potential inaccuracies from LLMs by reflectively verifying the validity of the generated vulnerability exploits, thus reducing the false alarm rate. As shown in Figure 4, its input includes the distilled prompt ①, the test program generated by the generator ③, and a basic instruction: *"Verify whether this test program can catch the vulnerability."* Through program execution and symbolic path analysis, the validator evaluates whether the test case successfully triggers the vulnerability. If the test case is effective, the validator outputs the result. If it fails, the failure information is relayed back to the generator, guiding further optimization and the next round of test case generation.

## 2.3 TEST EXECUTION

After generating vulnerability exploitation tests for downstream applications, VulFinder dynamically executes the test programs using testing frameworks such as JUnit or Pytest. This step further mitigates inaccuracies from LLMs and eliminates false positives. By leveraging the vulnerability description or executing the PoC program, users can obtain the program state that occurs when the vulnerability is triggered. This state is then compared with the execution result of the test program generated by VulFinder. If the two states align, it confirms that the vulnerability is reachable.

## 3 EXPERIMENTS

We conduct an empirical evaluation to assess the effectiveness and efficiency of VulFinder in detecting the reachability of vulnerabilities within SSC.

## 3.1 DATASETS

To evaluate the performance of VulFinder across different programming language ecosystems, this experiment constructed vulnerability datasets for Java and Python. For the Java dataset, we build based on Kang et al. Kang et al. (2022), which is a popular benchmark method, while Kang et al. provide an initial dataset and replica packages for their study, some downstream application data on GitHub are unavailable due to factors such as username changes, account deactivations, or library deletions. Additionally, the vulnerability fix submissions and validation procedures for several vulnerabilities are not disclosed in the released information, leading to further data gaps Chan & Chandy (2022). Consequently, we filtered out the invalid data from the Kang et al. dataset and expanded the dataset through manual collection, resulting in a final set of 16 Java vulnerabilities and 31 corresponding downstream software applications. In contrast, prior research has primarily focused on the Java ecosystem, and no publicly available vulnerability dataset exists for the Python ecosystem. To this end, we manually collect and organize data from 9 Python vulnerabilities and 19 associated downstream software applications, resulting in a total of 25 vulnerabilities spanning 17 software dependency packages and 50 downstream software applications with confirmed vulnerability reachability (positive dataset), as detailed in Table 1. Additionally, to evaluate VulFinder's ability to identify unreachable vulnerabilities, a negative dataset of 45 downstream software applications is compiled, including 24 Java applications and 21 Python applications with non-reachable vulnerabilities.

| Language | Vulnerabilities | Type | Packages | Downstream | TRANSFER | VESTA | VulFinder |
|---|---|---|---|---|---|---|---|
| Java | CODEC-134 Apache (2019) | Wrong functional behavior | Apache Codecs | 2 | 2 | 1 | 2 |
| | CVE-2020-13956 | Wrong functional behavior | Apache HttpClient | 2 | 2 | 2 | 1 |
| | HTTPCLIENT-1803 Snyk (2017) | Wrong functional behavior | Apache HttpClient | 2 | 0 | 0 | 0 |
| | CVE-2019-10094 | DoS | Apache Tika | 1 | 1 | 0 | 1 |
| | CVE-2020-28052 | Wrong functional behavior | Bouncy Castle | 2 | 2 | 1 | 2 |
| | CVE-2018-1000632 | XXE Injection | Dom4J | 2 | 1 | 1 | 2 |
| | CVE-2017-18349 | DoS | Fastjson | 2 | 0 | 0 | 1 |
| | CVE-2020-28491 | Out of memory | Jackson | 2 | 0 | 0 | 1 |
| | CVE-2020-15250 | Information leakage | JUnit | 2 | 2 | 0 | 2 |
| | CVE-2018-12418 | DoS | Junrar | 2 | 2 | 0 | 2 |
| | CVE-2020-5408 | Exception | Spring Framework | 2 | 2 | 2 | 2 |
| | CVE-2018-1274 | Out of memory | Spring Framework | 2 | 0 | 0 | 0 |
| | TwelveMonkeys-595 Snyk (2021a) | DoS | TwelveMonkeys | 2 | 0 | 0 | 2 |
| | CVE-2020-26217 | Remote code execution | XStream | 2 | 0 | 2 | 2 |
| | CVE-2017-7957 | Exception | XStream | 2 | 2 | 2 | 2 |
| | Zip4J-263 Snyk (2021b) | Exception | Zip4J | 2 | 2 | 2 | 2 |
| Python | CVE-2018-6188 | Information leakage | Django | 3 | 0 | – | 2 |
| | CVE-2018-7536 | DoS | Django | 1 | 0 | – | 1 |
| | CVE-2019-6975 | Out of memory | Django | 1 | 0 | – | 1 |
| | CVE-2018-1000656 | DoS | Flask | 1 | 0 | – | 0 |
| | CVE-2021-23437 | DoS | Python Pillow | 3 | 3 | – | 3 |
| | PyTorch-66946 Pytorch (2024) | Exception | PyTorch | 5 | 0 | – | 3 |
| | PyTorch-61656 | Exception | PyTorch | 1 | 0 | – | 1 |
| | PyTorch-54752 | Exception | PyTorch | 3 | 0 | – | 2 |
| | PyTorch-52822 | Exception | PyTorch | 1 | 0 | – | 1 |
| Total | 25 vulnerabilities | | 17 packages | 50 | 21 | 13 | 38 |

Table 1: The experimental vulnerability dataset includes vulnerable packages, vulnerability types, and the number of downstream reachable applications, along with the results of valid test program generation by VulFinder, VESTA and TRANSFER.

## 3.2 BASELINE

We compare the performance of the proposed VulFinder with the three state-of-the-art tools, including the currently used dependency analysis tool GitHub Dependabot, the popular exploit test generation tool TRANSFER, and the recently released VETSA, in assessing vulnerability reachability. We conduct experiments using the VESTA image released by Chen et al. (2024). It is worth noting that their tool is only applicable to Java, so the result of the Python dataset is none in Table 1. We also utilize the replication package for TRANSFER provided by Kang et al. (2022) to execute TRANSFER on Java datasets. Note that TRANSFER also does not include an implementation for the Python environment; for a comprehensive comparison with this popular baseline, we expand it to the Python-based replication tool, which is modeled on the implementation mechanism of TRANSFER. Specifically, we used Pynguin Lukasczyk et al. (2023) as the test case generation engine. However, although Pynguin is the currently the most mature test generation tool in the Python ecosystemBhatia et al. (2024), it only supports Python versions 3.8 to 3.10, while a large number of downstream software applications use newer or earlier Python versions, e.g., CVE-2018-6188 appeared before Django version 1.11.10, and the relevant version only supports Python 3.6 at the maximum. To ensure compatibility with these downstream application versions, this experiment includes modifications such as adjusting dependency versions and updating related API calls while maintaining the original functionality.

In addition, LLMs such as GPT-4o OpenAI (2024), GPT-3.5 OpenAI (2022), DeepSeek DeepSeek (2024), GLM-4 Bigmodel (2024), and Llama 3.3-70B Meta (2024) are employed as agents within VulFinder for comparison. AutoGen Wu et al. (2023) is used to facilitate interaction with the APIs of these LLMs and to orchestrate a multi-agent system consisting of a discriminator, a discriminator, a test generator, and a verifier.

## 3.3 EXPERIMENTAL SETUP

The experiment evaluates performance using three metrics: **Accuracy**, **Recall**, and **F1-score**, which are frequently used classification metrics. Accuracy and F1-score are used to evaluate the overall performance of classification, i.e., the tool performs in both the positive and negative classes. Recall measures the proportion of vulnerabilities correctly identified as reachable in the vulnerable positive

example dataset, and the closer the value is to 0, the higher the missing alarm rate of the tool. Using these performance metrics is a must for reference; however, we will also pay more attention to specific details when analyzing the results.

This experiment is conducted on a 3.7 GHz dual-core Intel Core i9 machine with 32 GB of RAM. Following the methodology of previous research Soltani et al. (2018), VESTA, TRANSFER, and VulFinder are executed 10 times for each dataset. If valid exploitable tests for a vulnerability are generated in at least 50% of the executions, the results are considered valid for the corresponding dataset.

## 3.4    Evaluation of VulFinder

We evaluate VulFinder's effectiveness and efficiency in verifying vulnerability reachability.

**Effectiveness Evaluation**   As shown in Table 2, overall, compared with the dependency-based analysis tool GitHub Dependabot and the most popular TRANSFER, VulFinder improves the Accuracy by 51% and 16%, and F1-score by 16% and 36%, respectively. Specifically, VulFinder improves the Accuracy by 45% and 7% and the F1-score by 15% and 14% on the Java dataset, respectively. And improves the Accuracy by 67% and 33% and the F1-score by 22% and 189% on the Python dataset, respectively. While the recently released VESTA method is only applicable to the Java language and is much smaller than VulFinder in all three metrics. This reflects the excellent effect of VulFinder on both positive and negative samples and its effectiveness on Java and Python programs in actual scenarios. In more detail, GitHub Dependabot's Recall reaches 1 and Accuracy is only 0.53, which means that it classifies all samples as positive, which highlights the inherent significant false positive rate caused by such methods not caring about the specific call relationship, that is, all reported negative classes are false positives. In contrast, tools such as VulFinder, VESTA, and TRANSFER, which incorporate dynamic validation with vulnerability exploit generation, have significantly improved Accuracy, indicating a reduction in false positives. However, the Recall of VESTA and TRANSFER is much lower than VulFinder, indicating that it has more missed reports and a narrower coverage of vulnerability diversity. Another evidence is that VulFinder is able to generate valid exploitable tests for 88% (22/25) of the vulnerabilities, while VESTA and TRANS-FER are only 32% (8/25) and 44% (11/25), respectively (see Table 1). This is because both tools rely heavily on manually defined rules, which limit their coverage and effectiveness. VulFinder's significant advantage shows that it significantly reduces false positives and false negatives, and improves accuracy and generalization. Furthermore, the performance of VulFinder is affected by the performance of the LLM used, but even so, it outperforms GitHub Dependabot and TRANSFER based on any one LLM and performs best when using GPT-4o.

| Tool | Dataset | Accuracy | | Recall | | F1-score | |
|---|---|---|---|---|---|---|---|
| GitHub Dependabot | Java | 0.56 | 0.53 | 1.00 | 1.00 | 0.72 | 0.69 |
| | Python | 0.48 | | 1.00 | | 0.64 | |
| VESTA | Java | 0.67 | – | 0.42 | – | 0.59 | – |
| | Python | – | | – | | – | |
| TRANSFER | Java | 0.76 | 0.69 | 0.58 | 0.42 | 0.73 | 0.59 |
| | Python | 0.60 | | 0.16 | | 0.27 | |
| VulFinder (GPT-4o) | Java | **0.81** | **0.80** | **0.77** | **0.76** | **0.83** | **0.80** |
| | Python | **0.80** | | **0.74** | | **0.78** | |
| VulFinder (GPT-3.5) | Java | 0.70 | 0.69 | 0.65 | 0.62 | 0.71 | 0.68 |
| | Python | 0.68 | | 0.58 | | 0.63 | |
| VulFinder (DeepSeek-V2.5) | Java | 0.78 | 0.78 | 0.74 | 0.74 | 0.79 | 0.78 |
| | Python | 0.78 | | 0.74 | | 0.76 | |
| VulFinder (GLM-4) | Java | 0.74 | 0.77 | 0.68 | 0.70 | 0.75 | 0.76 |
| | Python | 0.80 | | 0.74 | | 0.78 | |
| VulFinder (Llama-3.3) | Java | 0.76 | 0.77 | 0.71 | 0.70 | 0.77 | 0.76 |
| | Python | 0.78 | | 0.68 | | 0.74 | |

Table 2: Results of Effectiveness Analysis

**Efficiency Evaluation.**   VulFinder achieves an average efficiency improvement of over 1.5× compared to the best performance baseline TRANSFER. As shown in Table 3, three vulnerabilities

(HTTPCLIENT-1803, Zip4J-263, and CVE-2021-23437) are selected as case studies. Among these, Apache HttpClient and Zip4J are dependency packages from Java ecosystem, while Python Pillow is a package from Python ecosystem. The experimental results reveal that TRANSFER requires more time to generate exploit tests in the Python ecosystem compared to the Java dataset. For VulFinder, the time required to generate vulnerability exploit tests primarily depends on the inference time of the LLM and the network latency when calling the API. Since VulFinder iteratively processes function call paths during execution, its total generation time is linearly correlated with the length of these call paths. For instance, in the case of the HTTPCLIENT-1803 vulnerability, the downstream application *apache/gobblin* forms a call path with the vulnerable Apache HttpClient API by passing through dependencies four times Apach (2021), considerably increasing the time needed to generate an exploit test. Despite such factors, VulFinder demonstrates superior time efficiency and is less affected by variations in programming language environments, consistently outperforming TRANSFER in terms of time performance.

| Vulnerabilities | Packages | VulFinder | TRANSFER |
|---|---|---|---|
| HTTPCLIENT-1803 | Apache HttpClient | 23.2s | 33.5s |
| Zip4J-263 | Zip4J | 11.3s | 31.2s |
| CVE-2021-23437 | Python Pillow | 10.1s | 103.2s |

Table 3: Vulnerabilities Exploit Test Generation Time

## 3.5 ABLATION STUDY

We conduct ablation experiments to evaluate the role of each agent in the Multi-Agent mechanism, and the results are shown in Table 4. The Original setting represents the complete multi-agent driven framework, including the distillator, discriminator, generator, and validator. The other two settings involve removing the distillator in one case and removing both the discriminator and validator in the other.

| Settings | Dataset | Accuracy | | Recall | | F1-score | |
|---|---|---|---|---|---|---|---|
| Original | Java | 0.81 | 0.80 | 0.77 | 0.76 | 0.81 | 0.80 |
| | Python | 0.80 | | 0.74 | | 0.78 | |
| - distillator | Java | 0.69 | 0.66 | 0.61 | 0.58 | 0.69 | 0.65 |
| | Python | 0.63 | | 0.53 | | 0.57 | |
| - discriminator and validator | Java | 0.61 | 0.58 | 0.71 | 0.70 | 0.68 | 0.64 |
| | Python | 0.53 | | 0.68 | | 0.58 | |

Table 4: Results of Ablation Study

**Impact of distillator.** VulFinder employs a distillator to extract concise prompt statements from user input using the autoprompting mechanism. In our experiments, we remove the distillator and instead directly concatenate the raw user input to feed it into the subsequent multi-agent components for generating vulnerability test cases. The results reveal that removing the autoprompting mechanism causes overall Accuracy, Recall and F1-score to drop by 18%, 24%, and 19%, respectively. Specifically, causes Accuracy on the Java and Python datasets to drop by 15% and 21%, respectively, while Recall and F1-score decrease by 21% and 28%, 15% and 27%, respectively. These findings highlight the critical role of the autoprompting mechanism in enhancing VulFinder's performance. Raw user inputs often contain extensive contextual information, which can hinder the efficiency of LLMs in producing precise outputs Liu et al. (2024). While modern LLMs utilize attention mechanisms to process context Lyu et al. (2019), excessive contextual length can overwhelm the mechanism, making it difficult to capture long-distance dependencies and leading to inaccuracies in generated results. For instance, in the case of the PyTorch-52822 vulnerability, the user input included the vulnerability description, vulnerability patch information, downstream application code context, and the code block invoking the vulnerable API. The downstream application code context alone spanned over 500 lines of code facebookresearch (2024). Through its knowledge distillation process, VulFinder condenses such complex inputs into focused prompt statements, isolating the critical information needed to generate exploit tests effectively.

**Impact of discriminator and validator.** In the multi-agent mechanism, the discriminator evaluates the reachability of a dependency vulnerability and supplies relevant information to the generator, while the validator verifies the validity of the generated exploit test. As shown in Table 4, overall, removing the discriminator and validator resulted in 28% decrease in Accuracy, 9% decrease in Recall and 20% decrease in F1-score, indicating their critical role in accurately determining vulnerability reachability. Without discriminator and validator, VulFinder tends to generate vulnerability test programs indiscriminately, including for negative examples, leading to a proliferation of invalid test cases. This underscores the effectiveness of the discriminator and validator in filtering out non-triggerable cases and enhancing VulFinder's capability to produce valid exploit tests.

## 4 RELATED WORK

**Software Composition Analysis.** Software composition analysis tools such as OWASP Dependency-Check OWASP (2024) and commercially available SNYK Snyk (2024) are used to detect vulnerabilities in dependencies. However, studies across different software ecosystems have revealed these dependency analysis methods often produce false positives Alfadel et al. (2023); Decan et al. (2018). For instance, Elizalde Zapata et al. (2018) investigated the Node.js ecosystem and found that 73.3% of dependencies flagged as dangerous were actually safe. Similarly, Mir et al. (2023) observed that fewer than 1% of software projects had reachable call paths to the vulnerable code blocks. To address these issues, subsequent research has shifted toward finer-grained code-level analysis to assess dependency vulnerabilities. Some studies utilized static code analysis to generate static call graphs for software projects, determining the reachability of vulnerabilities based on graph connectivity Nielsen et al. (2021). However, discrepancies between static call graphs and actual runtime can lead to false positives. To overcome this limitation, dynamic code analysis approaches have been explored Foo et al. (2019), which generate call graphs and control flows by executing test cases. For example, Plate et al. (2015) analyzed whether vulnerable code blocks were actually executed by detecting call paths during runtime. Building on this, Ponta et al. (2018) proposed a hybrid approach combining static analysis and dynamic execution. This method, successfully applied in Java's Eclipse Steady tool, leverages the strengths of both approaches Ponta et al. (2020). However, these methods remain constrained by the limitations of test case coverage and the ability to trigger vulnerabilities under real-world conditions. Therefore, future research focuses on developing methods to generate vulnerability exploits, thereby verifying their reachability Iannone et al. (2021); Kang et al. (2022); Chen et al. (2024); Zhou et al. (2024).

**Vulnerability Exploit Test Generation.** The Vulnerability exploit is a program designed to verify the presence of a vulnerability. Brumley et al. (2008) introduced a method to automatically generate such tests using program patches. Xu et al. (2018) utilized symbolic execution and constraint solvers to create exploits for vulnerabilities in binary programs. However, these tools do not address dependency vulnerabilities within the SSC. Iannone et al. (2021) were the first to apply genetic algorithms to the task of generating vulnerability exploits, aiming to determine the reachability of vulnerability in SSC. Building on this foundation, Kang et al. (2022) enhanced the process by capturing the trigger conditions of vulnerabilities through the execution state of vulnerability PoC, significantly improving the efficiency of generating exploits for downstream applications. Chen et al. (2024) further optimized the approach by using genetic algorithms to transfer parameters from vulnerability PoC exploits to test programs for downstream applications, resulting in enhanced algorithmic efficiency. However, these methods fail to cover certain program mechanisms, leading to limitations in addressing specific vulnerabilities and are only applicable to the Java ecosystem.

## 5 CONCLUSION

This study introduces VulFinder, a multi-agent-driven test generation framework designed to verify the reachability of vulnerabilities within the SSC. VulFinder leverages the advanced comprehension capabilities of LLMs across various programming languages to overcome coverage limitations. By generating test cases along the call paths of vulnerable modules, VulFinder produces exploit tests for downstream applications and employs a multi-agent mechanism, incorporating discriminators and validators to assess and verify vulnerability reachability accurately. Experimental results show the superiority of VulFinder on both Java and Python datasets.

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
