# OpenReview forum: "VulFinder: A Multi-Agent-Driven Test Generation Framework for Guiding Vulnerability Reachability Analysis"
_ICLR.cc/2026/Conference — Submitted to ICLR 2026_

### Official Review · Reviewer_xRU1 · 2025-11-01

**Soundness:** 2
**Presentation:** 1
**Contribution:** 2
**Rating:** 2
**Confidence:** 4

**Summary:**

This paper proposes VulFinder, a multi-agent LLM-based system for assessing vulnerability reachability in software supply-chain dependencies. The system first constructs call paths from downstream projects to vulnerable APIs, then uses a distillator–discriminator–generator–validator loop to generate tests that allegedly verify whether the vulnerability is exploitable in practice. Experiments on Java and Python datasets show improvements over prior tools like TRANSFER and VESTA in accuracy and efficiency.

While the problem is important, the paper suffers from issues in presentation clarity, experimental rigor, and dataset transparency. Many methodological details, including how exploit success is judged, how attack surfaces are defined, and how ground truth is established, are not convincingly specified. The results feel anecdotal in places, and the evaluation omits key clarifications needed to trust the claims.

**Strengths:**

- Tackles an important and timely problem in software supply-chain security.
- Demonstrates cross-language evaluation (Java + Python).
- Multi-agent iterative refinement is a reasonable approach for LLM-guided test generation.
- Shows improvement over two recent baselines in reported metrics.
- Open-sourcing the tool is valuable for the community.

**Weaknesses:**

1. **Dataset construction and labeling lack clarity.**
The dataset is only partially inherited from prior work and otherwise curated manually, but the paper does not clearly explain how “confirmed reachability” was established.
2. **Attack-surface definition is oversimplified.**
The system treats any method on a call path to a vulnerable API as part of the attack surface. This ignores the distinction between internal calls and true external entrypoints such as CLI inputs or network endpoints. No taint analysis or input-flow tracking is used to ensure attacker control over parameters.
3. **Exploit success criteria are vague and limited to visible effects.**
The Validator primarily checks for observable runtime behavior, such as crashes or exceptions, but the paper does not define detection conditions, thresholds, or instrumentation. The ReDoS example (Figure 4) does not show how “performance degradation” is measured. Silent vulnerabilities (e.g., memory corruption, path traversal without exception) are not handled. Without explicit observability mechanisms, the evaluation favors vulnerabilities with obvious symptoms and may miss subtler classes.
4. **Evaluation reporting is selective and incomplete.**
Efficiency results (Table 3) are demonstrated using only three chosen vulnerabilities rather than reporting median or mean times across the dataset. This makes the performance claims anecdotal rather than statistically grounded.
5. **Presentation quality is below expectations.**
Important elements are poorly explained, such as color coding in Figure 1 and exploit-effect detection in Figure 4. Citation style is inconsistent (\cite vs \citep), and several methodological steps are described imprecisely. Critical concepts—attack surface, reachability definition, success criteria—are not formally defined.

**Questions:**

1. How exactly was vulnerability reachability “confirmed” in the positive dataset? Especially when the vulnerability does not produce observable side-effect?
2. How is the attack surface defined for downstream modules? How do you distinguish external entrypoints (e.g., main functions, http request handlers, etc.) from arbitrary internal public methods?
3. How does the Validator detect silent vulnerabilities (e.g., memory corruption, logic bugs without exceptions)?
4. For ReDoS (Figure 4), what concrete timeout / CPU usage criteria define “performance degradation”?
5. Why only report generation time for three chosen vulnerabilities? Can you provide dataset-wide averages and variance?
6. How do you handle dynamic behaviors (reflection, SPI, RPC entrypoints) beyond static call-graph matching?
7. Can you provide a breakdown of results by vulnerability category (e.g., DoS vs info-leak vs logic error)? Maybe classify them with CWEs?

**Details Of Ethics Concerns:**

The tool automatically generates exploit tests, which could be abused; responsible disclosure & usage framing needed.

---

### Official Review · Reviewer_uhbB · 2025-11-02

**Soundness:** 2
**Presentation:** 3
**Contribution:** 4
**Rating:** 6
**Confidence:** 3

**Summary:**

The paper introduces VulFinder, an agentic approach that generates exploit tests to determine if reported vulnerable APIs are reachable by the downstream application. VulFinder utilizes static analysis to generate call graphs leading to vulnerable APIs and multiple agents (a distillator, discriminator, generator, and validator) to perform auto-prompting, assessing vulnerability reachability, generating relevant test programs, and validating their effectiveness. VulFinder achieves 21% accuracy improvement over prior work VESTA, while achieves7% accuracy improvement over the baseline TRANSFER. VulFinder is demonstrated to work with both Java and Python, validating its generalizability.

**Strengths:**

+ A quick and reliable confirmation of whether reported vulnerabilities are affecting downstream applications provides developers with information to make a quick decision to mitigate or eliminate the vulnerability in their software.
+ The additional dataset would be useful for further research.

**Weaknesses:**

- The “SOTA” VESTA is not as effective as the baseline TRANSFER in the evaluation; however, in their paper, VESTA claims to be much more effective than TRANSFER. The discrepancy is not really explained, which leads to questions about the evaluation process.
- Additional information is needed for the process of curating the dataset. This would demonstrate the proper procedure of labeling and filtering, which in turn instills confidence in the evaluation result.
- The ablation study could also include an entry for VulFinder without the distillator, the discriminator, and the validator (essentially direct generation from call graphs). This is important to demonstrate the contribution of all additional components and compare VulFinder against a vanilla LLM.
- A qualitative investigation of the mode of failure would help future researchers better direct their efforts and make the study more complete.

**Questions:**

* Given that  VESTA is worse than TRANSFER,  why is this considered an SOTA approach? Also, in their paper, VESTA evaluation claims a much better effectiveness when compared to TRANSFER. What is the cause of this discrepancy?
* Do you have some insight into some of the cases that VulFinder failed to verify reachability? How would the different components contribute to these modes of failure? Would certain types of failure appear when certain components get removed?
* It seems LLM’s quality does affect the result. Would a more modern and effective LLM in coding tasks, such as Claude Sonnet would provide better results? Would a better foundation model close the gap between VulFinder and vanilla prompting (i.e., generator only)?
* Could you provide more details on how the datasets are built (Python dataset and additional Java entries)? How are the vulnerabilities selected, how are the downstream applications selected, and how are the labels of affected and not affected assigned?

---

### Official Review · Reviewer_95DV · 2025-11-03

**Soundness:** 2
**Presentation:** 1
**Contribution:** 2
**Rating:** 2
**Confidence:** 4

**Summary:**

The paper presents VulFinder, a multi-agent framework designed to more accurately determine whether software projects are affected by vulnerabilities in third-party components. Combining static code analysis with the code understanding abilities of LLMs, VulFinder uses agents to iteratively generate and execute exploit tests that validate vulnerability reachability. Experiments show that VulFinder significantly outperforms existing tools, improving accuracy and efficiency while reducing false positives and false negatives across multiple programming languages.

**Strengths:**

1. Addresses a well-defined and practical problem of vulnerability reachability analysis. Existing approaches, including dependency-based ones like Github Dependabot and heuristic-based ones like SIEGE and TRANSFER, both have limitations in terms of false positives and/or negatives that the proposed approach aims to overcome.

2. The emprical evaluation is reasonably comprehensive, targeting two constructed datasets one for Java and one for Python, for a total of 25 vulnerabilities across 17 packages and 50 downstream projects.  Baselines include Dependabot, SIEGE, and TRANSFER.

**Weaknesses:**

1. The presentation of the multi-agent driven test generation framework needs significant improvement. It is hard to understand what each of the different components (distillator, discriminator, generator, validator) does and how they interact. More generally, the paper does not explain what is really challenging about this problem, and what is truly novel about the architecture.

2. I did not see baselines of standalone state-of-the-art LLMs like the latest GPT reasoning model which would also help with better motivating a multi-agent architecture.

3. While the empirical results are promising, with VulFinder (GPT-4o) achieving high accuracy, recall, and F1 scores on the dataset, the paper would be significantly strengthened if the approach was applied on a larger test set in the wild.

4. Another way to convince a skeptical reader of the significance of the challenge being addressed would be to show the length of the call chains discovered. I did not see this metric in the experimental results but I could have overlooked it.

**Questions:**

Please see weaknesses.

---

### Official Review · Reviewer_DjbS · 2025-11-11

**Soundness:** 2
**Presentation:** 2
**Contribution:** 2
**Rating:** 2
**Confidence:** 4

**Summary:**

This paper proposes an agent-based system for vulnerability reachability analysis.

**Strengths:**

The proposed system can perform slightly better than traditional methods. However, deploying such an agent-based system to replace traditional approaches may incur high costs (see questions).

**Weaknesses:**

- The paper does not clearly demonstrate the benefits of using multi-agent design. The logic and control flow of the multi-agent system appear simple. The proposed method seems achievable with a single react-based agent with proper prompts or existing coding agents (e.g., Claude Code, Codex) given appropriate task descriptions. The authors could strengthen their contribution by comparing their multi-agent design with existing agents.
- The self-constructed benchmark includes only a limited set of tasks, which is insufficient to demonstrate the system's effectiveness or improvement over baselines. In addition, the authors claim cross-language generalizability, but the evaluation only covers two programming languages.

**Questions:**

What is the cost of running the agents? Can the authors provide a cost comparison between their approach and traditional methods?

---

### Meta-Review · Area_Chair_eMDc · 2026-01-09

**Summary:**

Reviewers agreed the paper targets an important and timely problem (vulnerability reachability analysis), but raised consistent concerns that the submission does not meet the bar for acceptance due to insufficient clarity and rigor. Multiple reviewers found the multi-agent design under-motivated, noting the system appears achievable via a simpler single-agent or strong standalone LLM baseline, and that the paper does not convincingly isolate what each agent contributes. Reviewers also questioned the trustworthiness and transparency of the evaluation, including limited dataset scale, unclear dataset construction and labeling procedures, selective efficiency reporting, and vague definitions of attack surface and exploit success criteria. In addition, concerns were raised about non-anonymous artifacts (GitHub link revealing identity) and a potentially problematic reference flagged by the program chairs, which further undermined the submission’s standing and presentation quality. Overall, while promising, the work was viewed as not yet sufficiently substantiated for acceptance.

**Reviewer Concerns:**

The rebuttal helped clarify the authors’ intent and provided some high-level justification for design choices and evaluation setup, partially addressing questions around motivation and cross-language evaluation. However, the key issues remained largely outstanding: reviewers still lacked confidence in the dataset construction / ground-truth confirmation process, the formal definition of reachability and exploit success, and the evaluation completeness, especially regarding attack-surface realism, observability for silent vulnerabilities, and broader in-the-wild validation. The rebuttal also did not fully resolve the central critique that the multi-agent architecture is insufficiently justified without stronger ablations and comparisons to modern standalone LLM baselines. Finally, the procedural concerns (non-anonymous GitHub link, and the flagged questionable citation) are not issues that rebuttal can meaningfully remedy during review and further contributed to the rejection decision.

**Reviewer Scores:**

Reviewer DjbS would likely remain at Reject (2), as their main concerns (limited dataset, unclear benefit of multi-agent design, and cost considerations) were not convincingly resolved. Reviewer 95DV would also likely stay at Reject (2), given persistent concerns about poor presentation clarity, lack of strong LLM-only baselines, and limited evidence of robustness at scale. Reviewer xRU1 would likely remain at Reject (2), as the rebuttal did not adequately address the most substantive methodological gaps (attack surface definition, success criteria, evaluation selectivity, and dataset transparency), and also flagged security/ethics framing issues. Reviewer uhbB, who was marginally positive, might move slightly downward (e.g., from 6 → 5) after seeing that other reviewers remained unconvinced and that the key methodological transparency issues (especially around dataset grounding and ablation completeness) were not fully closed. Overall, the discussion would likely strengthen consensus around rejection due to unresolved rigor and clarity issues.

---

### Decision · Program_Chairs · 2026-01-26

Reject